# Dose-rate coefficients for external exposure to radionuclides uniformly distributed in soil to an infinite depth

**Daiki Satoh**[1]*, **Nina Petoussi-Henss**[2]

**1** Nuclear Science and Engineering Center, Japan Atomic Energy Agency, Ibaraki, Japan, **2** External and Internal Dosimetry, Biokinetics, German Federal Office for Radiation Protection, Oberschleißheim, Germany

* satoh.daiki@jaea.go.jp

**Data Availability Statement:** All relevant data are within the manuscript and its Supporting Information files.

**Funding:** The authors received no specific funding for this work.

## Abstract

Using a database on external exposures to environmental sources provided by the International Commission on Radiological Protection, monoenergetic and nuclide-specific dose-rate coefficients have been evaluated for volumetric sources with a uniform distribution to an effectively infinite depth in soil. Organ equivalent and effective dose rates for the public (newborns; 1-, 5-, 10-, and 15-year-old children; and adults), ambient dose equivalent rates, and air kerma free-in-air rates at 1 m above the ground were computed. This was performed using the weighted-integral method for monoenergetic photon and electron sources in an energy region of $10^{-2}$ to 8 MeV with 25 energy points to obtain the respective monoenergetic dose-rate coefficients. Then, based on these data, the dose-rate coefficients for 1252 radionuclides of 97 elements were evaluated. In those computations, the dose contribution from bremsstrahlung generated by electrons in the soil was also considered. In addition, dose-rate coefficients for the primordial radioactive decay chains of the thorium, uranium, and actinium series, as well as the decay of $^{137}$Cs with $^{137m}$Ba in secular radioactive equilibrium, were obtained using the Bateman equation. For verification, the results of the effective dose rates for the $^{40}$K, $^{50}$V, thorium, and uranium series, as well as $^{137}$Cs/$^{137m}$Ba, were compared with those of previous studies and agreed within 10% for most cases. The results showed that the present dose-rate coefficients for radionuclides uniformly distributed to an infinite depth in soil were computed using appropriate procedures and can be used to assess external doses to the public, living on landfill soils containing naturally occurring radionuclides.

## Introduction

Nuclide-specific dose-rate coefficients have been evaluated to assess doses from external sources of radionuclides distributed in the environment [1–6]. The dose-rate coefficient is the dose-rate per unit radioactivity concentration of a radionuclide, and the dose can be evaluated if the radioactivity concentration is known. In 2020, the International Commission on Radiological Protection (ICRP) published a database of dose-rate coefficients for 1252 radionuclides of 97 elements distributed in soil, air, and water in ICRP Publication 144 [7]. This database

**Competing interests:** The authors have declared that no competing interests exist.

was the first of its kind issued by the ICRP. It included data on nuclide-specific coefficients for organ equivalent and effective dose rates for external exposures of the public (newborn; 1-, 5-, 10-, and 15-year olds; and adult males and females). The data were estimated following the ICRP 2007 recommendations and dosimetric methodology [8]. In addition, ambient dose equivalent and air kerma free-in-air rates (hereinafter referred to as "air kerma rates") 1 m above the ground were provided. These data have been welcomed by agencies in charge of assessing external doses from environmental radiation sources. In fact, they have been used by regulators in different European countries, the United Nations Scientific Committee on the Effects of Atomic Radiation (UNSCEAR) [9], and the United Kingdom Health Security Agency [10].

The nuclide-specific dose-rate coefficients of the ICRP are based on radiations emitted by the indicated radionuclide and do not consider radiations emitted by radioactive decay products, i.e., progeny. Moreover, the data of nuclides distributed in the soil as volumetric sources are given for eight exponential distributions, including the ground surface up to a depth of 100 g cm$^{-2}$. After the publication of the ICRP database in 2020, the demand for dose-rate coefficient data for radionuclides uniformly distributed in soil to an effectively infinite depth has increased, which is not addressed in ICRP Publication 144. The source condition is suitable for simulations of Naturally Occurring Radioactive Materials (NORMs), including $^{40}$K and progeny nuclides of $^{232}$Th and $^{238}$U decay series in the soil. This source configuration well approximates the distribution of radionuclides in landfill soils containing waste from industries involving NORMs, such as monazite mining. Exposure to NORM poses no real risk of a radiological emergency leading to tissue reactions or immediate danger to life because its radioactivity concentration is not high [11]. However, NORM is widely distributed and causes prolonged exposures. Actions against the prolonged exposures of workers and the public are within the scope of the radiation protection system where the principles of justification and optimization should apply [12].

Thus, the nuclide-specific dose-rate coefficients for volumetric sources uniformly distributed to an infinite depth in soil for 1252 nuclides of 97 elements were derived using the data of ICRP Publication 144. This study also evaluated the data of nuclides in secular radioactive equilibrium from $^{137}$Cs/$^{137m}$Ba, thorium, uranium, and actinium series and verified the values by comparing them with reported data. All the numerical data (organ, effective, air kerma, and ambient dose equivalent rates) are provided in the supplementary information (S1–S11 Tables).

## Materials and methods

This section first describes the dose coefficients for soil contamination provided in ICRP Publication 144 [7], which were employed to evaluate dose coefficients in this study, and the evaluation method used. Second, the method adopted in the present study to compute coefficients for uniformly distributed nuclides in soil to an effectively infinite depth is highlighted. Third, the treatment of radioactive decay chains is discussed.

### Database of ICRP Publication 144 for soil contamination

**Fig 1** shows a schematic representation of the simulated geometry for computing the dose-rate coefficients for soil contamination presented in ICRP Publication 144, and **Table 1** summarizes its specification. Radiation-transport simulations starting from widely distributed radiation sources in the environment require considerable computing time to achieve sufficient statistical accuracy of the radiation field at its center. It is inefficient to repeat the time-consuming simulation for every sex- and age-specific phantom. Therefore, in the methodology adopted for the work of ICRP Publication 144 and other similar studies [1–6], the radiation-

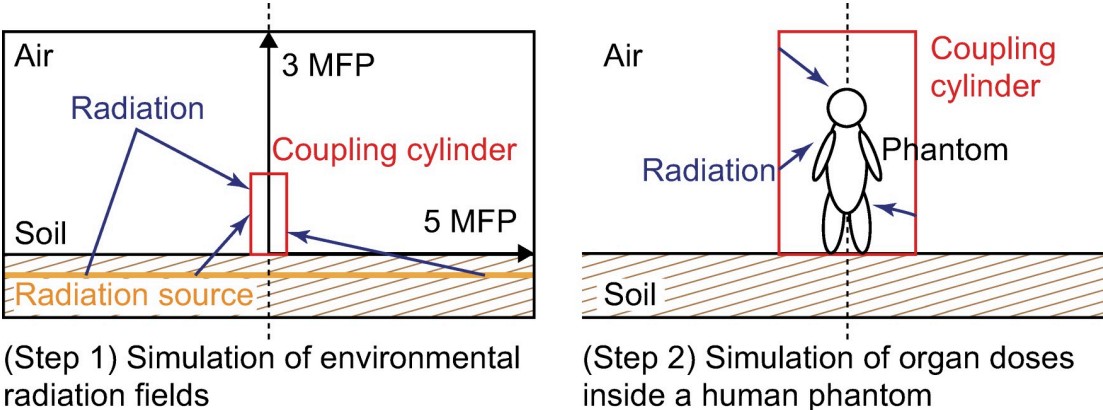

**Fig 1. Simulation steps for soil contamination following ICRP Publication 144.** MFP: mean-free path of photons in the air.

transport simulation was divided into two steps. (1) Simulation of the environmental radiation fields for monoenergetic photon and electron sources on and in the soil and (2) simulation of the doses absorbed by organs and tissues in the human body, i.e., reference phantoms exposed to environmental radiation. Therefore, the environmental radiation field obtained in Step 1 was used to the Step 2 simulation for the radiation transport in the phantoms.

In Step 1, planar sources of monoenergetic photons, whose energies ranged from $10^{-2}$ to 8 MeV with 25 energy points, were set to depths of 0.0, 0.2, 1.0, 2.5, and 4.0 mean-free path (MFP) of photons in the soil. Electron transport was only performed for a planar source of monoenergetic electrons on the ground surface as electrons in the soil hardly reach the surface. To model the environment that infinitely extends in the horizontal direction, the radius of the cylindrical simulation geometry was set to 5 MFP of photons in the atmosphere. The actual length was varied according to the source energy. The radiations propagating through the environment were recorded in terms of their type, energy, and angle of incidence on the surface of a virtual cylinder (60-cm diameter, 200-cm height; coupling cylinder) on the soil at the center of the geometry.

In Step 2, the radiation-transport simulation was restarted from the surface of the coupling cylinder in which air and the phantom were placed. Equivalent dose rates of organs and tissues, except for the skin, were calculated using the voxel-type ICRP reference pediatric (newborn, 1-, 5-, 10-, and 15-year-old children) [14] and adult [17] male and female phantoms. The skin doses were estimated using the mesh-type ICRP reference phantoms [15, 16, 18] for a 50-

**Table 1. Specifications of ICRP Publication 144 [7] database for soil contamination.**

| | |
|---|---|
| Radiation source | • Planar sources at depths of 0.0, 0.2, 1.0, 2.5, and 4.0 MFP[a];<br>• Planar sources at specific depths of 0.0, 0.5, 3.0, and 10.0 g cm$^{-2}$;<br>• Volumetric sources with exponential distributions for relaxation masses per unit area of 0.5, 1.0, 2.5, 5.0, 10.0, 20.0, 50.0, 100.0 g cm$^{-2}$ |
| Radiation type | Photon, electron, Bremsstrahlung |
| Energy range | $10^{-2}$–8 MeV |
| Radionuclide | 1252 nuclides of 97 elements listed in ICRP Publication 107 [13] |
| Phantom | ICRP reference newborn, 1-, 5-, 10-, 15-year old, adult, male and female [14–18] |
| Target dose | Organ equivalent dose rates, effective dose rates;<br>Ambient dose equivalent rate, air kerma rate, both at 1 m above the ground |

[a]MFP: mean-free path of photons in soil.

$\mu$m-thick layer within a depth range of 50–100 $\mu$m below the skin surface, which is considered to be the radiosensitive target layer. Effective dose rates were obtained based on the definition of the current ICRP system of radiological protection described in the ICRP 2007 Recommendations [8]. Therefore, the same set of tissue weighting factors was used in their computation for all age groups [7], and the differences in the effective dose rates as a function of age are only due to the anatomical differences in the phantoms. In addition, ambient dose equivalent and air kerma rates were computed at a height of 1 m above the ground. The combined relative uncertainties given as one standard deviation from both Step 1 and Step 2 simulations were less than 1% and 10%, respectively, for the air kerma rates and the equivalent dose rates of most organs and tissues.

The dose-rate coefficients for monoenergetic planar sources at depths in the soil expressed in MFP were evaluated using the aforementioned simulations. The unit of the dose-rate coefficients was nSv h$^{-1}$ Bq$^{-1}$ m$^2$, except for those of air kerma [nGy h$^{-1}$ Bq$^{-1}$ m$^2$]. Based on these data, the dose-rate coefficients for monoenergetic planar sources at specific depths of 0.0, 0.5, 3.0, and 10.0 g cm$^{-2}$ were computed and tabulated. Furthermore, the coefficients for volume sources with exponential distributions bearing relaxation masses per unit area of 0.5, 1.0, 2.5, 5.0, 10.0, 20.0, 50.0, and 100.0 g cm$^{-2}$ were computed and tabulated [7]. The distribution of the volumetric sources was limited to a depth of 100 g cm$^{-2}$ below the ground surface.

Nuclide-specific coefficients were deduced from the data for the monoenergetic sources according to the radioactive decay data of the 1252 nuclides of 97 elements presented in ICRP Publication 107 [13]. X-rays, gamma rays, annihilation photons, beta particles, and internal conversion and Auger electrons within the energy range of 10$^{-2}$ to 8 MeV were considered radiations emitted from the radioactive decay. The contribution of bremsstrahlung generated by beta particles and electrons in the soil was considered [1]. In ICRP Publication 144, dose-rate coefficients of a radionuclide only include the contribution of radiations from the decay of that nuclide and not the contributions from progeny nuclides produced in the decay chain.

## Dose-rate coefficients for volumetric sources in soil to an effectively infinite depth

First, the coefficients for photons, electrons, and bremsstrahlung were determined separately for monoenergetic volumetric sources in soil to an effectively infinite depth with 25 energy points from 10$^{-2}$ to 8 MeV. Thereafter, the coefficients were interpolated and summed, considering the radiations from each nuclide. For photons, the sources were uniformly distributed from the ground surface to the depth of 4 MFP in the soil; this depth can be regarded as an effectively infinite depth because the dose contribution to a person on the ground from deeper sources is negligible. The dose-rate coefficient of the infinite photon source, $\dot{d}_V^\gamma(E)$ [nSv h$^{-1}$ Bq$^{-1}$ kg], was obtained using the previously developed weighted-integral method [19, 20] as follows:

$$\dot{d}_V^\gamma(E) = \frac{1}{A} \int_0^{4MFP(E)} \dot{d}_P^\gamma(E, \zeta) a(\zeta) d\zeta \cdot L(E)\rho \tag{1}$$

where $E$ [MeV] is the energy of the monoenergetic photon source, and $A$ [Bq m$^{-1}$] and $a(\zeta)$ [Bq m$^{-2}$] are the total activity to the infinite depth and the activity concentration on the planar source at depth $\zeta$ [m], respectively. $\dot{d}_P^\gamma(E, \zeta)$ [nSv h$^{-1}$ Bq$^{-1}$ m$^2$] is the dose-rate coefficient of photons with energy $E$ from the planar source at depth $\zeta$. $L(E)$ [m] is the physical depth of the volumetric source obtained by converting 4 MFP of the photons in the soil with a density of $\rho$ [kg m$^{-3}$]. $a(\zeta)$ is a constant value with respect to $\zeta$ owing to its uniform distribution. The integration was numerically solved using a trapezoidal rule with small strips, in which the quantity,

$\dot{d}_P^\gamma(E, \zeta)$, was determined using the piecewise cubic Hermite interpolating polynomial (PCHIP) [21] for the data of ICRP Publication 144.

Only beta particles and electrons emitted from nuclides on the ground surface were considered; therefore, the dose-rate coefficient of beta particles and electrons for the infinite source, $\dot{d}_V^e(E)$ [nSv h$^{-1}$ Bq$^{-1}$ kg], was expressed as follows:

$$\dot{d}_V^e(E) = \frac{1}{A} \dot{d}_P^e(E, 0) a(0) \Delta\zeta \cdot L(E)\rho \tag{2}$$

where $\dot{d}_P^e(E, 0)$ [nSv h$^{-1}$ Bq$^{-1}$ m$^2$] is the dose-rate coefficient of an electron with energy $E$ at $\zeta$ = 0 from ICRP Publication 144. $a(0)$ [Bq m$^{-2}$] is the activity concentration of the source in a thin layer of the ground surface, whose thickness $\Delta\zeta$ was set to 0.1 $\mu$m. $A$ [Bq m$^{-1}$] is the total activity of the infinite source and has the same value as that in **Eq (1)**. Electrons emitted from an underground source cannot penetrate the soil; however, they produce bremsstrahlung, which is a type of photon. The maximum energy of the bremsstrahlung reaches the energy of the electron and can contribute to the dose rate above the ground. Using the same method as that in ICRP Publication 144, the dose contribution of bremsstrahlung from electron sources deeper than 0.5 g cm$^{-2}$ was considered. The dose-rate coefficients for monoenergetic bremsstrahlung, $\dot{d}_V^b(E)$ [nSv h$^{-1}$ Bq$^{-1}$ kg], were estimated using the data of planar photon sources, $\dot{d}_P^\gamma(E)$, as follows:

$$\dot{d}_V^b(E) = \frac{1}{A} \int_{B_{min}}^{4MFP(E)} \dot{d}_P^\gamma(E, \zeta) a(\zeta) d\zeta \cdot L(E)\rho \tag{3}$$

where $B_{min}$ is the lower integral limit corresponding to the physical depth of 0.5 g cm$^{-2}$ in the soil.

Nuclide-specific dose-rate coefficients for photon and electron sources, $\dot{d}_V^{N,\gamma}$ and $\dot{d}_V^{N,e}$ [nSv h$^{-1}$ Bq$^{-1}$ kg], respectively, were obtained from the results of the aforementioned monoenergetic photons and electrons using the following equations:

$$\dot{d}_V^{N,\gamma} = \sum_i Y_i^{N,\gamma}(E_i) \cdot \dot{d}_V^\gamma(E_i),$$

$$\dot{d}_V^{N,e} = \sum_i Y_i^{N,e}(E_i) \cdot \dot{d}_V^e(E_i) + \int_0^\infty \Phi^{N,e}(E) \cdot \dot{d}_V^e(E) dE \tag{4}$$

where $Y_i^{N,\gamma}(E_i)$ and $Y_i^{N,e}(E_i)$ are the yields of the photon and electron with discrete energy $E_i$ emitted from radionuclide $N$, respectively. $\Phi^{N,e}(E)$ [MeV$^{-1}$] in the integration of the second term for electrons denotes the yield of beta particles with a continuous energy spectrum. The data regarding radiation yields from the radioactive decay of nuclides were obtained from ICRP Publication 107 [13]. The dose-rate coefficients were estimated by interpolating the dose coefficients of monoenergetic sources (given for 25 energy points; 10$^{-2}$ to 8 MeV) using the PCHIP [21]. For bremsstrahlung, the nuclide-specific dose-rate coefficient, $\dot{d}_V^{N,b}$ [nSv h$^{-1}$ Bq$^{-1}$ kg], was evaluated as follows:

$$\dot{d}_V^{N,b} = \int_0^\infty \Phi^{N,b}(E) \cdot \dot{d}_V^b(E) dE \tag{5}$$

where $\Phi^{N,b}(E)$ [MeV$^{-1}$] is an energy fluence of bremsstrahlung produced by electrons emitted from the decay of nuclide $N$ in the soil. $E$ is the energy in a continuous bremsstrahlung

spectrum. The numerical data of $\Phi^{N,\mathrm{b}}(E)$ for each nuclide were obtained from a supplemental file of ICRP Publication 144, named "Brems_Spec. TXT".

Finally, the dose-rate coefficient of radionuclide $N$, $\dot{d}_V^N$ [nSv h$^{-1}$ Bq$^{-1}$ kg], was computed by summing the components of photons, electrons, and bremsstrahlung as follows:

$$\dot{d}_V^N = w^\gamma \dot{d}_V^{N,\gamma} + w^e \dot{d}_v^{N,e} + w^\gamma \dot{d}_V^{N,b} \tag{6}$$

where $w^\gamma$ and $w^e$ are the radiation weighting factors for photons and electrons, respectively, both with values of 1. $w^\gamma$ was applied for bremsstrahlung because it is a type of photon.

Using the aforementioned procedure, dose-rate coefficients for 1252 nuclides of 97 elements distributed in the soil to an infinite depth were obtained, including organ equivalent dose rates for males and females separately for newborns; 1-, 5-, 10-, and 15-year-olds; and adults. Furthermore, the respective effective dose rates; ambient dose equivalent rates; and air kerma rates at 1 m above the ground were obtained.

The present dose-rate coefficients were calculated for idealized and hypothetical exposure situations, for instance, for an unclothed reference human standing in an upright position on unpaved soil. In real exposure situations, radiations, particularly electrons, could be attenuated by pavement materials. However, similarly to previous studies [3, 4, 7], this effect was not considered in the present study.

## Radioactive decay chain in secular equilibrium

Fig 2 illustrates the decay chains of the thorium, uranium, and actinium series, which are in secular radioactive equilibrium, with the branching fractions from the parent nuclide to the daughters. The information on the radioactive decay chains was obtained from ICRP Publication 107 [13]. The thorium series, headed by $^{232}$Th, is the most abundant of all naturally occurring radionuclides, with a half-life of $1.405 \times 10^{10}$ years, and the uranium series, headed by $^{238}$U, has a half-life of $4.468 \times 10^9$ years. The actinium series is headed by $^{235}$U with a half-life of $7.04 \times 10^8$ years, which is less important compared with the other primordial decay chains owing to its abundance. The dose contribution of alpha particles from alpha decay is negligible because of their short range in the soil and was, therefore, not considered here for the evaluation of the dose-rate coefficients. In addition to the primordial radionuclide decay chains, the decay from $^{137}$Cs to $^{137\mathrm{m}}$Ba was considered. Notably, $^{137\mathrm{m}}$Ba, which emits a gamma ray of 0.662 MeV with a half-life of 2.552 min, is formed by the decay of $^{137}$Cs, whose half-life is 30.167 years, with a branching fraction of 0.944 and establishes a secular equilibrium.

Based on the Bateman equation [22, 23], the radioactivity of the $i$-th nuclide in the radioactive decay chain, $A_i(t)$, at time $t$ is as follows:

$$A_i(t) = A_1(0) \prod_{j=1}^{i-1} \left( f_{j,j+1} \cdot \lambda_{j+1} \right) \sum_{j=1}^{i} \frac{\exp\left( -\lambda_j t \right)}{\prod_{\substack{k=1 \\ k \neq j}}^{i} \left( \lambda_k - \lambda_j \right)} \tag{7}$$

where $j$ and $k$ are the members of the decay chain, $A_1(0)$ is the activity of the 1-st nuclide at $t = 0$, $\lambda_j$ is the decay constant for the $j$-th radionuclide, and $f_{j,j+1}$ is the branching fraction of the nuclear decay of chain member $j$ forming member $(j + 1)$. Using the activity, $A_i(t)$, the dose, $D$,

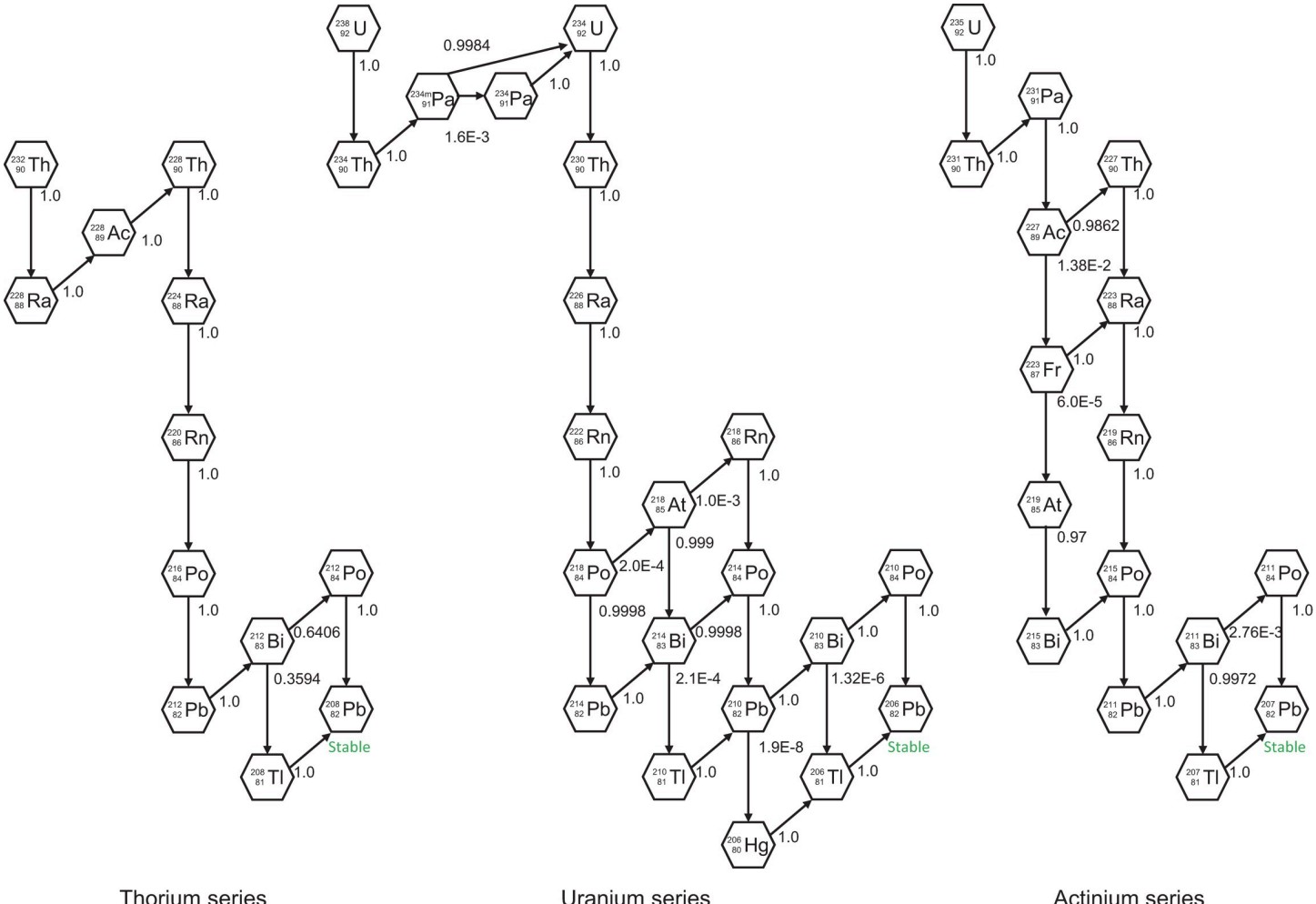

**Fig 2. Decay chains of the thorium, uranium, and actinium series in secular equilibrium treated.** The numerical values indicate the blanching fraction of the decay from one nuclide to another taken from ICRP Publication 107 [13].

during the period of $t = 0$ to $t = T$, considering the decay chain of the radionuclide, is as follows:

$$D = \int_0^T \sum_{i=1}^n A_i(t) \cdot \dot{d}_v^i dt,$$

$$= A_1(0) \sum_{i=1}^n \dot{d}_v^i \prod_{j=1}^{i-1} \left( f_{j,j+1} \cdot \lambda_{j+1} \right) \sum_{j=1}^i \frac{1 - \exp\left(-\lambda_j T\right)}{\lambda_j \prod_{\substack{k=1 \\ k \neq j}}^i \left( \lambda_k - \lambda_j \right)} \tag{8}$$

where $n$ is the number of nuclides in the decay chain, and $\dot{d}_v^i$ is the dose-rate coefficient of the $i$-th nuclide evaluated in this study. When secular equilibrium was established, **Eq (8)** was

simplified as follows:

$$D = A_1(0) \frac{1 - \exp(-\lambda_1 T)}{\lambda_1} \cdot \sum_{i=1}^{n} \dot{d}_V^i \prod_{j=1}^{i-1} f_{j,j+1} \tag{9}$$

Using Eq (9), the dose-rate coefficient of the radioactive decay chain, $C$, in secular equilibrium, $\dot{d}_{s.e.}^C$ [nSv h$^{-1}$ Bq$^{-1}$ kg], was as follows:

$$\dot{d}_{s.e.}^C = \sum_{i=1}^{n} \dot{d}_V^i \prod_{j=1}^{i-1} f_{j,j+1} \tag{10}$$

For example, the dose-rate coefficient of the decay from $^{137}$Cs to $^{137m}$Ba, $\dot{d}_{s.e.}^{Cs-137/Ba-137m}$ [nSv h$^{-1}$ Bq$^{-1}$ kg], was computed as follows:

$$\dot{d}_{s.e.}^{Cs-137/Ba-137m} = \dot{d}_V^{Cs-137} + 0.944 \dot{d}_V^{Ba-137m} \tag{11}$$

where $\dot{d}_V^{Cs-137}$ and $\dot{d}_V^{Ba-137m}$ are the dose-rate coefficients of $^{137}$Cs and $^{137m}$Ba, respectively, given by Eq (6), and 0.944 is the branching fraction of $^{137}$Cs forming $^{137m}$Ba.

## Results and discussion

This section presents the results of the effective dose-rate coefficients for monoenergetic photon and electron sources uniformly distributed to an effective infinite depth. In addition, their age and energy dependencies are discussed along with the relationship between ambient dose equivalent and air kerma rates. The dose-rate coefficients for specific radionuclides and decay chains in secular equilibrium are provided, and the results are compared with those of previous studies [3, 4]. All numerical data, including the results of the organ equivalent dose rates for 1252 nuclides of 97 elements, are provided in the supplementary information (S1–S11 Tables).

### Dose-rate coefficients for monoenergetic radiation sources

Fig 3 shows the effective dose-rate coefficients for monoenergetic photon sources, together with the coefficients of the ambient dose equivalent rate, $\dot{h}^*(10)$ and air kerma rate, $\dot{k}_a$, 1 m above ground. The age-dependent effective dose rates were higher for the younger age groups because children with shorter stature have organs and tissues closer to the soil, i.e., to the source. The operational quantity, $\dot{h}^*(10)$ representing the equivalent dose rate at a depth of 1 cm from the surface of a tissue-equivalent sphere [8], exhibited larger values than the effective dose rate, which is the protection quantity, for the entire energy range, except at 0.01 MeV. Note that $\dot{h}^*(10)$ has limited practical application in radiological protection for low-energy photons around 0.01 MeV [24]. $\dot{k}_a$ was observed to be larger than the effective dose rate for the entire energy range. The results showed that the effective dose rate to the public from photon-emitting radionuclides uniformly distributed to an infinite depth in the soil can be adequately controlled by monitoring the ambient dose equivalent and air kerma rates.

Fig 4 shows the effective dose-rate coefficients for monoenergetic electrons, with the coefficients of the ambient dose equivalent rate, $\dot{h}^*(10)$, and air kerma rate, $\dot{k}_a$, for monoenergetic photon sources. As mentioned, the total activity of the volumetric electron source was calculated assuming a uniform distribution to a depth equivalent to 4 MFP of photons in the soil, as in the case of photon sources. However, only the electron sources in the 0.1-$\mu$m thick top layer of the ground surface contributed to the dose rates above the ground. Consequently, the dose-

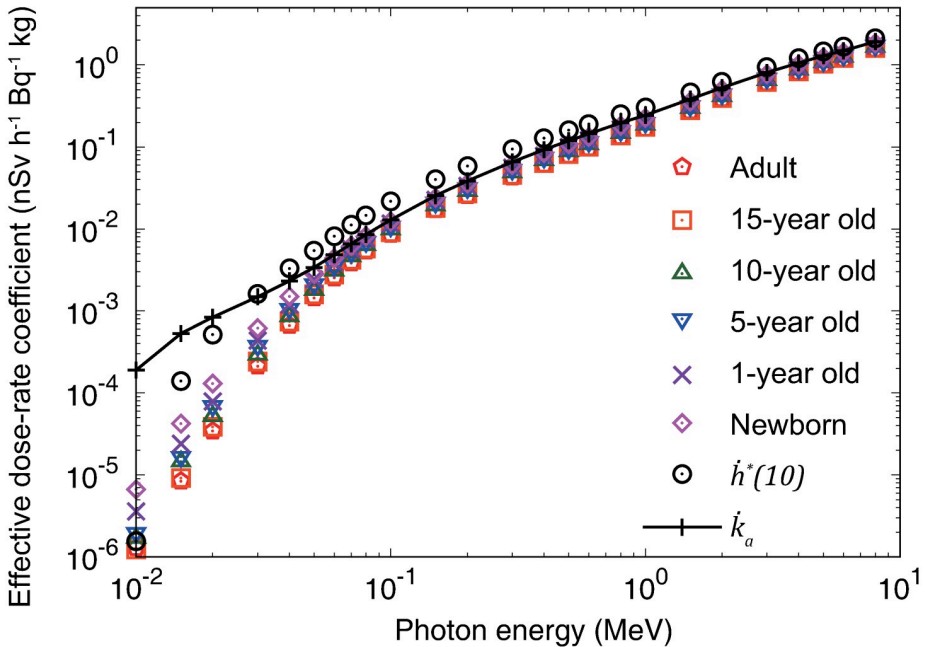

**Fig 3. Effective dose-rate coefficients for monoenergetic photon sources uniformly distributed to an infinite depth in the soil and corresponding ambient dose equivalent rate, $\dot{h}^{*}(10)$, and air kerma rate, $\dot{k}_a$, 1 m aboveground.** Numerical data are provided in S12 Table.

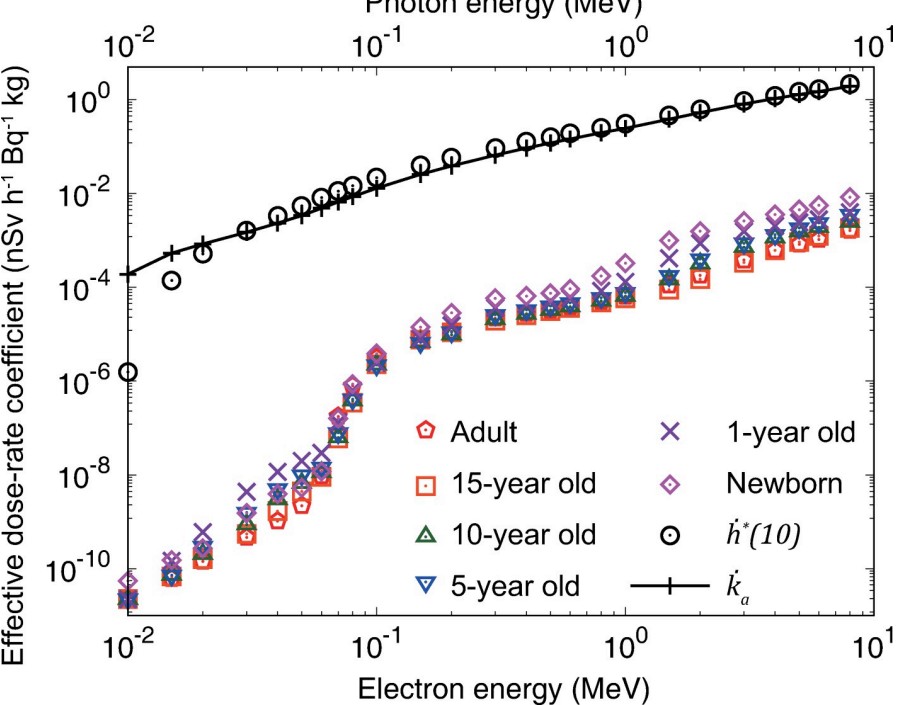

**Fig 4. Age-dependent effective dose-rate coefficients for monoenergetic electron sources uniformly distributed to an infinite depth in the soil.** The open circles and solid line with cross marks indicate the coefficients of ambient dose equivalent and air kerma rates, respectively, for the photon sources, which are the same as those shown in Fig 3. Numerical data are provided in S13 Table.

rate coefficients of electron sources normalized to the total radioactivity were more than one order of magnitude lower than those of the photon sources. The effective dose rates to electron sources were mainly contributed by the skin equivalent dose rates, calculated for the sensitive 50–100 $\mu$m layer below the skin surface of the mesh-type ICRP reference phantoms [15, 16, 18]. Electrons with an energy less than 0.06 MeV cannot reach the sensitive layer of the skin and deposit energy by the secondary photons, including bremsstrahlung produced in the body.

## Nuclide-specific dose-rate coefficients

Table 2 lists the nuclide-specific coefficients of age-dependent effective dose, ambient dose equivalent, and air kerma rates for typical nuclides in NORM exposures. These nuclides have a long half-life of over a billion years and do not form a radioactive decay chain. Notably, $^{40}$K is one of the main sources of natural radiation on the earth because of its abundance. This nuclide undergoes nuclear transformations into $^{40}$Ar and $^{40}$Ca, releasing a gamma ray of 1.46 MeV with a branching fraction of 0.109 and a beta-minus particle, whose maximum energy is 1.31 MeV, with a branching fraction of 0.891. The abundances of the other nuclides were not so significant. However, the dose-rate coefficients of $^{176}$Lu were of the same order of magnitude as those of $^{40}$K because the average energies of gamma rays and beta particles emitted via a single nuclear transformation are similar. For $^{50}$V and $^{138}$La, the average photon energy per nuclear transformation exceeds 1 MeV, and these coefficients were higher than those of the other nuclides. $^{87}$Rb and $^{115}$In are pure beta emitters, and $^{123}$Te is an X-ray emitter. Thus, these nuclides exhibit low dose-rate coefficients.

Table 3 lists the dose-rate coefficients for radionuclides of the thorium, uranium, and actinium series, as well as $^{137}$Cs/$^{137m}$Ba in secular radioactive equilibrium. To verify the computational procedure applied in the present study, the results were compared with those of previous studies [3, 4] calculated with similar source and phantom configurations. Table 4 provides a comparison of the effective dose-rate coefficients derived in the present study for adults, 10- and 5-year-old children, and newborns with values of previous studies. The data of the $^{232}$Th series, $^{238}$U series, and $^{40}$K were obtained from Petoussi-Henss et al. [4], who evaluated the dose rates for homogeneous distributions up to a depth of 100 cm in the soil using the ICRP voxel adult phantoms [17] and the Helmholtz Zentrum München (HMGU) voxel pediatric phantoms [25] in conjunction with the radiation-transport simulation code, EGSnrc [26]. The dose coefficients for $^{137}$Cs/$^{137m}$Ba were derived by Eq (11) from the respective data of $^{137}$Cs and $^{137m}$Ba from Bellamy et al. [3], who employed stylized hermaphrodite phantoms [27] and the simulation code, MCNP6 [28]. Their data were also used for the $^{40}$K and $^{50}$V.

**Table 2. Effective dose-rate [nSv h$^{-1}$ Bq$^{-1}$ kg]; ambient dose equivalent rate, $\dot{h}^*(10)$ [nSv h$^{-1}$ Bq$^{-1}$ kg]; and air kerma rate, $\dot{k}_a$ [nGy h$^{-1}$ Bq$^{-1}$ kg], coefficients for nuclides uniformly distributed to an infinite depth in the soil.** $\dot{h}^*(10)$ and $\dot{k}_a$ were estimated at 1 m above the ground. Here, 15-y, 10-y, 5-y, and 1-y indicate 15-, 10-, 5-, and 1-year-old children, respectively.

| Nuclide | Effective dose-rate coefficient (nSv h−1 Bq−1 kg) | | | | | | $\dot{h}^*(10)$ | $\dot{k}_a$ |
| | Adult | 15-y | 10-y | 5-y | 1-y | Newborn | | |
| --- | --- | --- | --- | --- | --- | --- | --- | --- |
| $^{40}$K | 2.86E−02 | 2.95E−02 | 3.10E−02 | 3.33E−02 | 3.49E−02 | 3.79E−02 | 4.86E−02 | 4.01E−02 |
| $^{50}$V | 2.58E−01 | 2.67E−01 | 2.81E−01 | 3.01E−01 | 3.16E−01 | 3.40E−01 | 4.40E−01 | 3.64E−01 |
| $^{87}$Rb | 7.85E−06 | 7.93E−06 | 8.07E−06 | 9.17E−06 | 1.14E−05 | 1.69E−05 | 7.82E−06 | 4.91E−06 |
| $^{115}$In | 1.62E−05 | 1.68E−05 | 1.83E−05 | 2.18E−05 | 2.69E−05 | 3.71E−05 | 2.08E−05 | 1.31E−05 |
| $^{123}$Te | 1.69E−07 | 1.92E−07 | 2.32E−07 | 3.14E−07 | 3.59E−07 | 5.30E−07 | 1.55E−06 | 1.61E−06 |
| $^{138}$La | 2.16E−01 | 2.24E−01 | 2.36E−01 | 2.53E−01 | 2.66E−01 | 2.87E−01 | 3.77E−01 | 3.08E−01 |
| $^{176}$Lu | 6.34E−02 | 6.71E−02 | 7.17E−02 | 7.86E−02 | 8.32E−02 | 8.95E−02 | 1.42E−01 | 9.69E−02 |

**Table 3. Effective dose-rate [nSv h⁻¹ Bq⁻¹ kg]; ambient dose equivalent rate, $\dot{h}^*(10)$ [nSv h⁻¹ Bq⁻¹ kg]; and air kerma rate, $\dot{k}_a$ [nGy h⁻¹ Bq⁻¹ kg], coefficients for nuclides of the radioactive decay chain in secular equilibrium uniformly distributed to an infinite depth in the soil.** Decay chains headed by ²³²Th, ²³⁸U, and ²³⁵U are the thorium, uranium, and actinium series shown in Fig 2, respectively. The ¹³⁷Cs chain indicates the secular radioactive equilibrium between ¹³⁷Cs and ¹³⁷ᵐBa.

| Nuclide | Effective dose-rate coefficient (nSv h−1 Bq−1 kg) | | | | | | $\dot{h}^*(10)$ | $\dot{k}_a$ |
|---|---|---|---|---|---|---|---|---|
| | Adult | 15-y | 10-y | 5-y | 1-y | Newborn | | |
| ²³²Th series | 4.10E-01 | 4.25E-01 | 4.48E-01 | 4.80E-01 | 5.04E-01 | 5.43E-01 | 7.20E-01 | 5.82E-01 |
| ²³⁸U series | 3.05E-01 | 3.16E-01 | 3.33E-01 | 3.59E-01 | 3.77E-01 | 4.07E-01 | 5.48E-01 | 4.37E-01 |
| ²³⁵U series | 8.25E-02 | 8.71E-02 | 9.30E-02 | 1.02E-01 | 1.08E-01 | 1.16E-01 | 1.84E-01 | 1.27E-01 |
| ¹³⁷Cs/ ¹³⁷ᵐBa | 9.19E-02 | 9.50E-02 | 1.01E-01 | 1.09E-01 | 1.15E-01 | 1.25E-01 | 1.77E-01 | 1.37E-01 |

The relative deviations between the data of the present work and those of Petoussi-Henss et al. [4], who used voxel phantoms, are less than 4%. The deviations to the data of Bellamy et al. [3], who used stylized hermaphrodite phantoms for the calculations, were found to be within 10% for ¹³⁷Cs/¹³⁷ᵐBa and ⁵⁰V; note that for these nuclides, photons play a dominant role in the dose contribution. The equivalent dose rates of skin, muscle, and skeleton were slightly higher than those of the other organs and tissues; however, there are no organs or tissues that exhibit markedly higher dose rates than other organs. It was also found that the deviations between the present and previous results do not vary significantly among age groups.

For ⁴⁰K which releases a beta-minus particle per nuclear transformation with a branching fraction of 0.891, the deviations of the present results to the data of Bellamy et al. were up to approximately 30%. Both studies employed the same methodology [1] for the generation of bremsstrahlung by the electrons inside the soil and their contribution to organ equivalent dose rates. However, the methodology for the dose contribution to skin by the primary electrons released near the ground surface was quite different. Bellamy et al. used the point-kernel method [29] for calculating the electron skin dose, while the present work employed the results of radiation-transport simulations for electrons and photons. Since the deviations are less than 10% for nuclides for which photons are the major contributor to the dose rate, it can be concluded that the disagreement is due to the difference in methodology for estimating skin equivalent dose rate from primary electrons.

## Conclusions

A database of monoenergetic and nuclide-specific dose-rate coefficients for volumetric sources in soil with a uniform distribution to an effectively infinite depth has been developed. The

**Table 4. Ratio of the effective dose-rate coefficients reported by Petoussi-Henss et al. [4] and Bellamy et al. [3] for adults, 10- and 5-year-old children, and newborns to those obtained in the present work.** Here, 10-y and 5-y indicate 10- and 5-year-old children, respectively.

| Nuclide | Ratio of effective dose-rate coefficient | | | |
|---|---|---|---|---|
| | Adult | 10-y | 5-y | Newborn |
| ²³²Th series | 1.04ᵃ | 1.00ᵃ | 1.01ᵃ | 1.03ᵃ |
| ²³⁸U series | 1.04ᵃ | 1.01ᵃ | 1.00ᵃ | 1.03ᵃ |
| ¹³⁷Cs/¹³⁷ᵐBa | 1.08ᵇ | 1.08ᵇ | 1.05ᵇ | 1.07ᵇ |
| ⁴⁰K | 1.04ᵃ | 1.01ᵃ | 1.00ᵃ | 1.02ᵃ |
| | 1.27ᵇ | 1.29ᵇ | 1.25ᵇ | 1.26ᵇ |
| ⁵⁰V | 1.07ᵇ | 1.08ᵇ | 1.05ᵇ | 1.06ᵇ |

ᵃDerived from the data of Petoussi-Henss et al. [4].
ᵇDerived from the data of Bellamy et al. [3].

coefficients were evaluated based on the data of ICRP Publication 144 for organ equivalent and effective dose rates for members of the public (newborns; 1-, 5-, 10-, and 15-year-old children; and adults), and ambient dose equivalent, and air kerma rates at 1 m above ground. We considered the dose contributions by photons, electrons, and bremsstrahlung from the sources within the energy region of $10^{-2}$ to 8 MeV. The effective dose rates increased with a decrease in the age of the subjects, and they were conservatively controlled by the ambient dose equivalent and air kerma rates in that energy region, except at 0.01 MeV. From the results of the monoenergetic sources, the nuclide-specific coefficients were computed for 1252 radionuclides of 97 elements presented in ICRP Publication 107. In addition, coefficients for the thorium, uranium, and actinium series, as well as $^{137}$Cs/$^{137m}$Ba in secular radioactive equilibrium, were obtained. The results of certain typical nuclides were compared with those of previous studies. A good agreement within 10% was observed, except for $^{40}$K which undergoes beta-minus decay with branching fraction of 0.891. The present data for $^{40}$K, which were derived from the results of explicit electron and photon transport simulations, exhibited approximately 30% lower values than those of previous calculation [3] employing the point-kernel method for the estimation of skin equivalent dose rate by electrons.

The present results based on the latest dose coefficients, decay data, and phantoms of the ICRP enable the assessment of doses against prolonged exposures of the public who live on soils containing naturally occurring radionuclides uniformly distributed to an infinite depth. Although it is necessary to consider effects, such as shielding from soil pavement materials and buildings and the population living habits, i.e., location and occupancy factors, the present dose-rate coefficients are very useful in demonstrating compliance with the dose limits and considering justification and optimization in radiation protection systems. In addition, the data could contribute to a risk assessment and management for the use of waste from NORM industries by considering both non-radiological and radiological hazards. All the data are provided in the supplementary information (S1–S11 Tables).

## Supporting information

**S1 Table. Effective dose rate coefficients for nuclides uniformly distributed to an infinite depth in the soil.**
(TXT)

**S2 Table. Organ equivalent dose rate coefficients for female phantoms and nuclides uniformly distributed to an infinite depth in the soil.**
(TXT)

**S3 Table. Organ equivalent dose rate coefficients for male phantoms and nuclides uniformly distributed to an infinite depth in the soil.**
(TXT)

**S4 Table. Organ equivalent dose rate coefficients for female phantoms and $^{232}$Th series uniformly distributed to an infinite depth in the soil.**
(TXT)

**S5 Table. Organ equivalent dose rate coefficients for male phantoms and $^{232}$Th series uniformly distributed to an infinite depth in the soil.**
(TXT)

**S6 Table. Organ equivalent dose rate coefficients for female phantoms and $^{238}$U series uniformly distributed to an infinite depth in the soil.**
(TXT)

**S7 Table. Organ equivalent dose rate coefficients for male phantoms and $^{238}$U series uniformly distributed to an infinite depth in the soil.**
(TXT)

**S8 Table. Organ equivalent dose rate coefficients for female phantoms and $^{235}$U series uniformly distributed to an infinite depth in the soil.**
(TXT)

**S9 Table. Organ equivalent dose rate coefficients for male phantoms and $^{235}$U series uniformly distributed to an infinite depth in the soil.**
(TXT)

**S10 Table. Organ equivalent dose rate coefficients for female phantoms and $^{137}$Cs/$^{137m}$Ba uniformly distributed to an infinite depth in the soil.**
(TXT)

**S11 Table. Organ equivalent dose rate coefficients for male phantoms and $^{137}$Cs/$^{137m}$Ba uniformly distributed to an infinite depth in the soil.**
(TXT)

**S12 Table. Numerical values plotted in Fig 3.**
(TXT)

**S13 Table. Numerical values plotted in Fig 4.**
(TXT)

## Acknowledgments

We express our gratitude to Dr. K. Saito of the Japan Atomic Energy Agency (JAEA) for fruitful discussions related to external exposures to radiations from naturally occurring radionuclides in soil. We also thank Dr. A. Endo of the JAEA for his advice on using the radioactive decay data of the ICRP Publication 107.

## Author Contributions

**Conceptualization:** Daiki Satoh, Nina Petoussi-Henss.

**Data curation:** Daiki Satoh.

**Investigation:** Daiki Satoh, Nina Petoussi-Henss.

**Methodology:** Daiki Satoh, Nina Petoussi-Henss.

**Project administration:** Daiki Satoh.

**Validation:** Daiki Satoh.

**Writing – original draft:** Daiki Satoh.

**Writing – review & editing:** Nina Petoussi-Henss.

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
