## [Decision Letter · Decision Letter 0]

17 May 2024

PONE-D-24-10704Dose-rate coefficients for external exposure to radionuclides uniformly distributed in soil to an infinite depthPLOS ONE

Dear Dr. Satoh,

Thank you for submitting your manuscript to PLOS ONE. After careful consideration, we feel that it has merit but does not fully meet PLOS ONE’s publication criteria as it currently stands. Therefore, we invite you to submit a revised version of the manuscript that addresses the points raised during the review process.

 Please submit your revised manuscript by Jul 01 2024 11:59PM. If you will need more time than this to complete your revisions, please reply to this message or contact the journal office at plosone@plos.org. Please include the following items when submitting your revised manuscript:A rebuttal letter that responds to each point raised by the academic editor and reviewer(s). You should upload this letter as a separate file labeled 'Response to Reviewers'.A marked-up copy of your manuscript that highlights changes made to the original version. You should upload this as a separate file labeled 'Revised Manuscript with Track Changes'.An unmarked version of your revised paper without tracked changes. You should upload this as a separate file labeled 'Manuscript'.If applicable, we recommend that you deposit your laboratory protocols in protocols.io to enhance the reproducibility of your results. Protocols.io assigns your protocol its own identifier (DOI) so that it can be cited independently in the future. For instructions see: https://journals.plos.org/plosone/s/submission-guidelines#loc-laboratory-protocols. Additionally, PLOS ONE offers an option for publishing peer-reviewed Lab Protocol articles, which describe protocols hosted on protocols.io. Read more information on sharing protocols at https://plos.org/protocols?utm_medium=editorial-email&utm_source=authorletters&utm_campaign=protocols.

We look forward to receiving your revised manuscript.

Kind regards,

Aaron Specht

Academic Editor

PLOS ONE

Journal Requirements:

**Additional Editor Comments:**

Please address each reviewer in a point by point response and make changes to the manuscript accordingly.

Reviewers' comments:

Reviewer's Responses to Questions

**Comments to the Author**

1. Is the manuscript technically sound, and do the data support the conclusions?

Reviewer #1: Yes

Reviewer #2: Yes

2. Has the statistical analysis been performed appropriately and rigorously? 

Reviewer #1: Yes

Reviewer #2: Yes

3. Have the authors made all data underlying the findings in their manuscript fully available?

Reviewer #1: Yes

Reviewer #2: Yes

4. Is the manuscript presented in an intelligible fashion and written in standard English?

Reviewer #1: Yes

Reviewer #2: Yes

5. Review Comments to the Author

Reviewer #1: This study tabulates radionuclide-specific external dose coefficients as effective dose rate, ambient dose rate, and air kerma rates for reference age-specific ICRP reference mesh-type phantoms. Where ICRP 144 tabulated dose coefficients for soil contamination located closer to surface, this work tabulates dose coefficients from infinite soil contamination depth. The manuscript is well-articulated and contains appropriate explanation of the theory and methods implemented. The following recommendations are requested to improve the rigor of the manuscript and to provide expansion of necessary details required the work conducted.

1. The authors have conducted a limited comparison to prior work.

a) A short discussion is requested on the impact of specific organ dose contributions to the deviation comparing the ICRP reference and stylized phantoms (i.e., which organs most profoundly affected the effective dose calculation).

b) A selected comparison of specific age groups is also requested to determine the provenance of variability between phantom type vs. source generation methods

2. The authors are requested to cite the original methods for the tabulation of the infinite soil dose coefficients, with discussion of comparison of methodology (beyond phantom) regarding source terms (e.g., comparison of bremsstrahlung calculated from both studies) from DOI:https://doi.org/10.1007/s00411-017-0692-7

3. The authors are requested to note the convergence criteria (e.g., 5% relative uncertainty) for the organs and tissue from the Monte Carlo simulation.

4. While the authors have listed prior work having addressed soil dose coefficients, while only addressing surface soil, the following work might be cited as another work that employs the discussed method: https://doi.org/10.1007/s00411-019-00812-2

Reviewer #2: Thank you for the opportunity to review this manuscript! This is a fantastic study to address concerns of naturally occurring radioactive material. There are some issues I would like to discuss and have address prior to publication.

In the introduction section you state, “Exposure to NORM poses no real risk of a radiological emergency leading to tissue reactions or immediate danger to life.” Additionally, throughout the article you again refer to the risk. Please further expand on the risk associated with NORM. It would be helpful for the reader to have a comparison to other radiation sources or other non-radiation hazards.

In the results and discussion section, “The age-dependent effective dose rates were higher for the younger age groups because children with shorter stature have organs and tissues closer to the soil, i.e., to the source.” Younger age groups have a higher sensitivity to radiation. Please address radiation sensitivity varies between an adult and a child. Based on your results, the risk associated to all age groups appears to be negligible.

Thank you again! I thought you did a great job to address the topic!

6. PLOS authors have the option to publish the peer review history of their article (what does this mean?). If published, this will include your full peer review and any attached files.

Reviewer #1: No

Reviewer #2: No

---

## [Author Response · Author response to Decision Letter 0]

9 Aug 2024

We have incorporated all the comments and suggestions. You can find our responses to each point in a rebuttal letter.

---

## [Decision Letter · Decision Letter 1]

3 Sep 2024

Dose-rate coefficients for external exposure to radionuclides uniformly distributed in soil to an infinite depth

PONE-D-24-10704R1

Dear Dr. Satoh,

We’re pleased to inform you that your manuscript has been judged scientifically suitable for publication and will be formally accepted for publication once it meets all outstanding technical requirements.

Kind regards,

Aaron Specht

Academic Editor

PLOS ONE

---

## [Editor Report · Acceptance letter]

17 Sep 2024

PONE-D-24-10704R1 

PLOS ONE

Dear Dr. Satoh, 

I'm pleased to inform you that your manuscript has been deemed suitable for publication in PLOS ONE. Congratulations! Your manuscript is now being handed over to our production team.

Kind regards, 

on behalf of

Dr. Aaron Specht 

Academic Editor

PLOS ONE